# Review of White Line Disorders in Zone 3 and Toe Tip Necrosis in Dairy Cows and Recent Insights into Aetiopathogenesis and Treatments

**DOI:** 10.3390/microorganisms13092159

**Published:** 2025-09-16

**Authors:** Menno Holzhauer, Han de Leeuw

**Affiliations:** 1Royal GD Animal Health, P.O. Box 9, 7400 AA Deventer, The Netherlands; 2Private Claw Trimmer, 1562 GP Krommenie, The Netherlands; j.leeuw986@upcmail.nl

**Keywords:** claw disorders, claw horn disruption, toe tip necrosis, correct use of antibiotics, bacterial

## Abstract

White line disorders represent the most prevalent claw horn disruption lesion in dairy cattle. Recent studies have yielded new insights into the appropriate treatment modalities for these lesions. The aims of this study are to elucidate the pathogenesis of white line disorders and its associated claw lesions, such as toe tip necrosis, and to discuss practical treatment applications. In Western Europe, many herds are endemically infected with digital dermatitis. White line disorders in zone 3 and toe tip necrosis starting in zone 1—often beginning as axial white line lesions—frequently exhibit a suboptimal response to standard treatments, including corrective trimming, the application of a hoof block on the healthy claw and the administration of NSAIDs, due to secondary infections with *Treponema* spp. This study addresses the current perspectives on the aetiopathogenesis of white line disorders and the therapeutic challenges in promoting complete recovery and the correct use of antibiotics, along with preventive measures, such as good flooring. An important factor of its pathogenesis is a decrease in body condition around parturition, Correct diagnosis can be made via the use of regular locomotion scoring and good diagnostic tools, and thin soles by among others overtrimming should be prevented. Current therapeutic methods consist of the prompt application of a block and an NSAID and, in some circumstances, a parenteral injection with antibiotics when there is no good response to the applied therapies.

## 1. Introduction

Lameness in dairy cattle, alongside mastitis and decreased fertility, represents a significant herd health issue. Due to the severe pain and prolonged duration associated with most claw disorders, lameness adversely impacts animal welfare and leads to substantial economic losses. These economic impacts primarily stem from reduced milk production, premature culling, the loss of body weight before slaughter, and increased labour requirements [1,2,3,4,5,6]. Furthermore, the estimated prevalence of lameness is high, affecting approximately 30% of cows in the Netherlands, and managing this condition is time-consuming, with additional labour often yielding suboptimal results. Due to less flexibility at the moment of lying down and standing up in cubicles, claw lesions are also frequently associated with hock lesions and lesions in the carpal region [7]. A Danish investigation concluded that locomotor disorders are responsible for approximately 40% of all euthanised cows, making them the most frequent cause of on-farm euthanasia in dairy cows [8]. Recently, the benchmarking of claw health has been introduced, enabling the comparison of individual herd claw health with that of numerous other dairy farms exhibiting similar performance levels. This benchmarking may further support analyses of the improvement potential of farm herds, encourage collaboration between farmers and veterinarians to enhance animal welfare, and assist in minimising economic losses due to lameness [9]. Additionally, maintaining good claw health can contribute to increased job satisfaction among farmers.

The primary cause of lameness in dairy cattle originates predominantly from the hoof and surrounding tissues, and it can be of either infectious or non-infectious origin. Infectious claw disorders are typically associated with effects on the skin around the hoof and in the interdigital space. The most frequently observed infectious lesions include digital dermatitis (DD) and interdigital phlegmon (IP). Digital dermatitis is associated with infections by bacteria such as *Treponema* spp., while interdigital phlegmon is commonly linked to *Fusobacterium necrophorum*. Additionally, bacteria such as *Dichelobacter nodosus*, *Porphyromonas levii*, and *Prevotella melaninogenica* are frequently found in IP lesions [10,11]. The most frequently noted non-infectious claw disorders, also known as claw horn disruption lesions (CHDLs), include white line disorders in zone 3 (WLDs), sole ulcers (SUs), and toe tip necrosis in zone 1 (TTN) (zone classification according to van der Tol et al., 2002 [12]), which have prevalence rates of approximately 18%, 9%, and 2–3%, respectively, in The Netherlands [13]. Similar prevalence data have been estimated in other Western European countries, such as the United Kingdom and Switzerland [14,15], as well as in Canada (Alberta) [16]. The aetiopathogenesis of CHDLs is multifactorial and often related to sole horn thickness, sole horn moisture content [17], and management practices, including housing conditions, bedding, exercise patterns (e.g., sharp turning around corners), and grooved floors [14,16,17]. Other frequently mentioned risk factors for CHDLs are: the sinking of the pedal bone [18] around parturition; a decrease in claw horn cushion thickness as a consequence of a decrease in total body condition [19,20,21]; fatty liver-related diseases, such as mastitis and endometritis; nutrition (e.g., barley grain, protein, and fibre); housing and feeding management; calving; season; age; growth rate; genetics; conformation; and behaviour [22,23,24,25]. Nutritional deficiencies, particularly of trace elements and the vitamin biotin, have also been implicated in the development of CHDLs, as confirmed by longitudinal field studies [26,27]. These studies demonstrated that, among cattle with laminitis-related CHDLs, the survival rate was higher in cows supplemented daily with biotin (20 mg/kg).

About fifteen years ago, a study conducted in Liverpool identified an association between *Treponema* bacteria involved in DD and three “non-healing” claw horn lesions: TTN, “non-healing white line disorders” (NH-WLDs), and non-healing sole ulcers. *Treponema* bacteria were identified in these types of lesions in DD-infected herds; in regular infections, pyrogenic bacteria such as *Arcanobaterium pyogenes* (formerly called *Trueperella pyogenes*) are usually identified. These lesions are characterised by penetration through the horn and most probably subsequent infection of the corium by *Treponema* bacteria [28]. This association was later corroborated by studies on cattle with similar “non-healing” horn lesions in Austria [29] and Switzerland [30], as well as in various investigations involving goats with severe lameness issues in the UK and Germany [31,32].

The objective of this contribution is to discuss current perspectives on the aetiopathogenesis of and therapeutic approaches for DD-associated WLD abscesses in zone 3 and toe necrosis in zone 1 in dairy cattle, as presented in the ICAR Claw Health Atlas Appendix A (January 2020), including the correct use of antibiotics. Although, new insights into these aspects of sole ulcers, these were excluded from this paper.

## 2. Materials and Methods

### Scope of the Papers Examined in This Review

Articles from countries with modern dairy production systems were selected, where lameness was related to claw horn lesions and more specific to WLD in zone 3 and TTN in zone 1. A preliminary database search for scientific articles (described below) returned 43 papers related to the pathogenesis of these disorders and 6 papers related to their treatment, with nearly all being from Western countries. This comprehensive literature search was conducted in 2025 to identify relevant studies involving white line and toe necrosis lesions in dairy cows. The search terms included “white line”, “pathogenesis”, or “treatment”, and “dairy cows” in the title, in the abstract, and/or as a keyword. The listed search terms were entered into three databases:

Web of Science PubMed and ScienceDirect. To perform this systematic review in a reasonable period of time, the publication period was set to the last two decennia, that is, from 2000 to 2024. This period was chosen because new insights into the aetiopathogenesis of CHDLs have been obtained since that time, as well as new insights into the correct treatment of CHDLs, which has recently changed. Thus, the outcome parameters of interest in these publications were the aetiopathogenesis and treatment of WLD and TTN, which are frequently related to each other [33].

A total of 121 papers were retrieved from the electronic database search for pathogenesis-related studies, and the first selection of potentially interesting papers was made (see Figure 1). Among these papers, 39 (33.9%) contained pathogenesis investigations, and, of these, the majority (16/39 = 41.0%) investigated heritability. The estimated h^2^ in almost all of the papers was between 0.06 and 0.15 for TTN and CHDLs, respectively [34]. The excluded papers contained prevalence studies, estimations of the costs of claw disorders (including white line disorders), or analyses of other bovine diseases or disorders. Table 1 presents the selected papers, along with the first author, year of publication, no. of cows included, study design, journal of publication, and the subject of the study. The main results of these studies are presented in Appendix A. Among them, 15 (38%) were case-control studies, and 15 (38%) were cross-sectional studies; regarding those remaining, 3 (8%) were observational cohort studies, 2 (5%) were prospective confirmation studies, 2 were prospective observational studies, 1 was a retrospective cohort study, and 1 was a retrospective confirmation study. Furthermore, 22 (56%) were published in the *Journal of Dairy Science*, 5 were published in the *British Veterinary Journal*, and 3 were published in the *Canadian Veterinary Journal*; regarding those remaining, 2 were published in *Animals*, 2 were published in *Veterinary Record*, 1 was published in *Veterinary Pathology*, 1 was published in the *New Zealand Veterinary Journal*, 1 was published in in *Veterinary and Animal Science*, 1 was published in *Research and Veterinary Science*, 1 was published in the *Journal of Animal Breeding and Genetics*, and 1 was published in *Tropical Animal Health and Production*.

## 3. Pathogenesis

As stated above, the development of WLD in zone 3 and TTN in zone 1 as part of the CHDL complex is considered to be multifactorial. Both WLD and TTN are part of the bovine laminitis complex, and they have been found to originate from a combination of heritability (h^2^: 0.05–0.15) [34], nutrition [68], and housing management [69]. As previously mentioned, factors mainly related to nutrition are the sinking of the pedal bone [18] and the loss of condition around parturition [19,70], and good rumen function and good mineral supply are important [71,72]. Regarding housing, factors such as pasturing when possible, cubicles with a good size and bedding, the prevention of overcrowding, and the use of rubber flooring are important [70,73]. The most widely accepted explanation for the development of TTN in feedlot cattle is probably “abrasion theory”, which posits that it is caused by variations in the hardness and elasticity [23,74] and, consequently, increasing moisture content of the solar horn and the thinning of the soles, as well as indirectly by white line separation [17]. Excessive wear of the solar horn leads to separation along the apical portion of the white line, which likely allows for secondary bacterial infections. In areas endemic for DD in herds, infections with *Treponema* spp. may penetrate the corium and progress to P3 osteitis, P2 osteomyelitis, tendonitis, tenosynovitis, cellulitis, and, in some cases, septicaemia [28]. In the absence of DD, infections with pyrogenic bacteria may cause problems. According to Jelinsky et al., this theory explains the main risk factors for TTN [75]. Although Western Europe does not have feedlot cattle, the same problems are also seen in dairy herds endemically infected with DD with proven *Treponema* infections [28]; the prevalence of WLD has doubled compared to that 20 years ago (9% vs. 18%, respectively) [13,76], and the prevalence of TTN varies from 1 to 3% and even up to 10% [77]. To the best of our knowledge, comparable infections with *Treponema* spp. are not seen in herds without DD. In line with a Canadian study, our experience is that TTN starts with a disorder or separation of the axial white line (zone 1), allowing for secondary bacterial infection [48,78]. In dairy cattle, CHDLs are related to certain genetic factors, greater production and ration [34,71,72], and housing factors, such as grooved slats, sharp turns, and a high stocking density. Research has shown that CHDL rates are lower in cattle kept in free stalls [79] and on rubber-covered slatted floors than in those kept on uncovered slatted and concrete floors [80,81,82]. Furthermore, attention should be paid to overtrimming by unexperienced claw trimmers [83].

## 4. Treatment

After observing a lame cow, identifying which leg is the most probable cause of lameness, and determining that the cause originates from the claw, the cow is brought into the trimming chute, and, normally, one starts with the trimming of the affected foot. It is advised to start with the seven steps of prophylactic claw trimming, as developed about fifty years ago by Toussaint Raven [84]. After completing these steps, the horn shoe is checked for a pain response with a hoof tester to determine the location of the WLD lesion, thereby allowing for a conclusion to be drawn as to whether a white line disorder is the most probable cause of lameness [85]. Many people such as those from the Nottingham Research Group and the Faculty of Veterinary Medicine in Malaysia have found a positive effect with the use of a wooden block on the contralateral claw of the same leg (the inner claw of the hindleg in most cases) and the parenteral application of an NSAID [66,86]. So, the next step is correctly applying the block before the therapeutic trimming of the affected outer claw. The goal of the block is to reduce the weight on the diseased claw. This approach only covers cases where the infection is limited to the dermis and not those with deep digital sepsis. In the case of sepsis, the parenteral application of antibiotics should be considered to limit the consequences of the infection (e.g., to prevent the joints from being affected) [87].

### 4.1. White Line Disorder in Zone 3

After all these preparations, one has to make the correct diagnosis, therapeutically trim the affected claw, and treat the cow using the following steps:
Examination and Diagnosis:Sometimes, the exact location of the lesion is not clear, and, in such cases, it is advisable to use a hoof tester to locate the specific area of pain and separation.The affected hoof should be carefully investigated to determine the extent of the damage and to ensure that there is no deeper infection. This allows for the differentiation between a white line fissure and a white line abscess.Trimming and Cleaning:The affected area of the hoof should be trimmed by removing any loose or damaged horn and exposing the affected area. Sometimes, part of the hoof wall has to be removed to allow the lesion to be relieved of pus and pressure from the corium, which also relieves the cow of pain. In the case of WLD type I (see Figure 1), this treatment on the affected hoof is usually sufficient. Besides the cleaning of the separated area to thoroughly remove any debris, dirt, or manure that might be trapped in the crack, no additional topical application of any therapeutic agent has been proven to be helpful. However, the parenteral application of an NSAID is helpful [66,83,86].Figure 1White line disorder grade 1, affecting a small area of the dermis, according to the ICAR Claw Health Atlas.
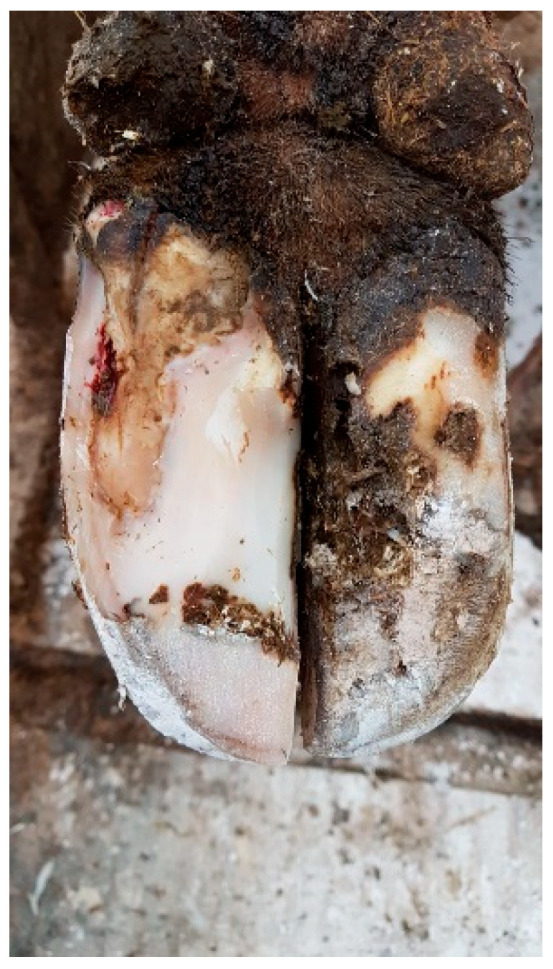
Debridement:If the local infection is more extensive (“non-healing white line abscesses”, ICAR Claw Health Atlas), it may be necessary to remove more of the claw horn, and this often becomes complicated. Thus, if standard treatment does not improve or cure the lesion, then it is advisable to let this step be performed by an experienced foot trimmer or veterinarian with the support of anaesthesia.All necrotic (dead and contaminated) tissue and any foreign material that might be present should be removed. Again, this may involve careful removal of the hoof wall to ensure that all damaged tissue is removed. This surgical debridement should always be performed using local anaesthesia. This is a time-consuming process that is usually not carried out at the same time as regular claw trimming of the whole herd (see Figure 2A–D). For topical treatment, one can consider products based on copper and zinc chelate or salicylic acid and methyl salicylate; other disinfecting products might also be recommended.Figure 2Presentations of cows with serious non-healing white line disorders at zone 3 before (**A**,**C**) and 3 months after (**B**,**D**) topical treatment and a parenteral injection with tilmycosin (10 mg/kg BW. SC). We also used in these cases hard plastic blocks beneath the inner claw, that hardly do not wear.
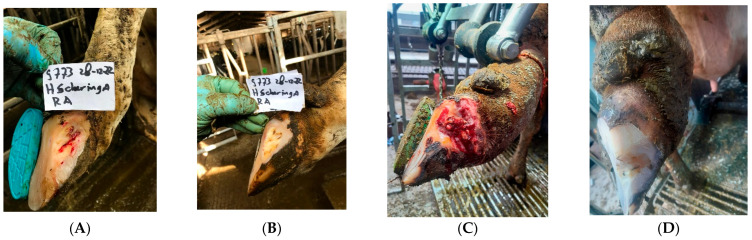
NSAIDs and AntibioticsIf repeated remedial trimming with topical treatment does not result in clinical cure, a single parenteral antibiotic treatment (e.g., 10 mg/kg BW. SC of tilmycosin) can help to achieve complete clinical cure, and most cows will be suitable for further milk production [67].

### 4.2. Toe Tip Necrosis (See Figure 3)

Examination and DiagnosisThe toe tip should be carefully inspected to determine the extent of the damage, and the depth of the infection should be determined with a probe. Samples should be sent to a laboratory for further investigation. It is not worthwhile to only send in a swab or horn sample for bacteriology, and Polymerase Chain Reaction (PCR) for *Treponema* is only performed in a small no. of laboratories in Europe. Infections may penetrate the corium and progress to P3 osteitis; P2 osteomyelitis; tendonitis; tenosynovitis; cellulitis (caused by *E. coli* or *Arcanobaterium pyogenes*); and, in some cases, septicaemia, which leads to an embolic event that culminates in death [31]. According to the sparse literature on the background of TTN, a low intake of both selenium and magnesium may play a role [68].Trimming and cleaningIn terms of TTN, different therapeutic options are now available, as solely claw trimming and the topical application of a product such as tetracycline powder or spray are usually not sufficient and also undesirable with the risk of developing antibiotic resistance. More effective options are as follows:
Loco-regional/regional anaesthesia and the removal of the entire affected horn and necrotic bone tissue with a grinder, as proposed by Kofler [88];The removal of necrotic tissue using local anaesthesia and surgical intervention or amputation of the tip of the claw [89];Loco-regional/regional anaesthesia and the removal of the entire affected horn and necrotic bone tissue, in combination with topical application of a disinfectant non-antibiotic unguent and a single parenteral antibiotic treatment, e.g., tilmycosin [67].


**Figure 3 microorganisms-13-02159-f003:**
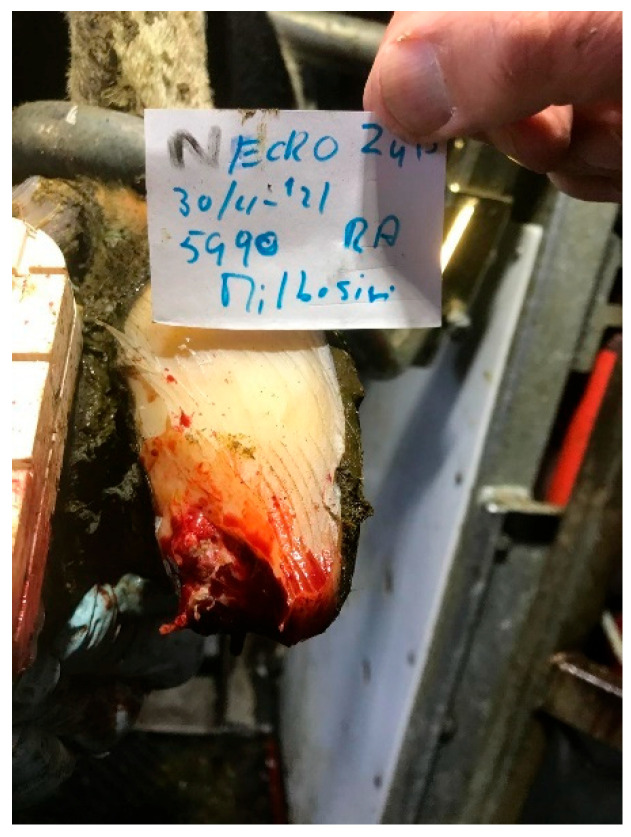
Serious toe tip necrosis.

All options should be combined with correct pain management such as a block under the claw (the inner claw in most cases) and the parenteral application of an NSAID. In a study performed in 2024, plastic foot blocks were used, which, unlike wooden blocks, do not wear unevenly.

## 5. Prevention

To prevent CHDLs and especially WLD lesions (both WLD in zone 3 and TTN in zone 1), it is necessary to perform regular locomotion scoring of the herd at two-week intervals in order to detect lameness and WLD lesions at an early stage [90], perhaps before the secondary infection of the corium by *Treponema* spp.; additionally, attention should be paid to proper preventive claw trimming, nutrition, and the environment. Proper claw trimming refers to the correct dorsal wall length; a sole horn thickness of at least 5–7 mm; the correct heel height, especially of the medial hind claw; and the proper removal of load from the claw affected by WLD separation or other CHDLs (step 4 of functional hoof trimming). Research from Sweden showed that, if CHDLs are the main cause of lameness, then strategic claw trimming (of lame cows, cows at the start of the dry standing period, and cows around 3 months in milk) is preferable to the trimming of all cows twice a year [91]. This advice is related to the greatest risk period (1–3 months pp.) of these disorders [92]. Good nutrition for the prevention of CHDLs is mainly based on good rumen function [71,72], which results in maximal endogenous biotin production, and a good supply of essential minerals such as zinc, copper, manganese, and selenium [68]. Good housing management for the prevention of CHDLs involves good cow comfort, which means the prevention of overcrowding, good adjustment of the cubicles, preferably the use of rubber-coated floors [73], and a clean and dry environment to prevent further contamination and to promote healing. Good cow comfort can be monitored using the cow standing index, as proposed by Cook et al. [93]. It is important to consult a bovine veterinarian to determine the appropriate course of treatment for white line disease in zone 3 in cattle. These practitioners can provide guidance and perform underfoot anaesthesia, and they may need to intervene if the condition is severe or if there are complications such as deep infections. This will hopefully prevent long-lasting lameness, improve cattle welfare and longevity, and enhance job satisfaction for dairy farmers and bovine practitioners.

## 6. Discussion

This review is limited to studies providing insights into the aetiopathogenesis and treatment of dairy cattle with white line lesions in zone 3 and toe tip necrosis conducted during the period of 2000–2025. This period was chosen because many studies conducted around the turn of the century examined the displacement of the pedal bone within the horn shoe, and those conducted more recently provided insights into the optimal treatment of claw horn lesions, which should contribute to the better welfare and longevity of dairy cows.

White line disorders in zone 3 are one of the most prevalent CHDLs in dairy cattle in Western Europe and the USA [14,70]. An investigation in the UK showed that increased parity, increased herd size, cows at pasture by day and housed at night, and solid grooved concrete floors in yards or alleys were the main risk factors for WLD in zone 3 [14], while a study from the USA focused on unbalanced weight bearing and metabolic, enzymatic, and hormonal changes [94]. Our experience is that WLD in zone 3 and TTN, which almost always starts as an axial WLD, are part of the bovine laminitis complex and a consequence of a combination of factors such as: the sinking of the pedal bone [18]; a decrease in body condition around the dry period [53,95]; poor horn quality; and sharp turns in walkways, which is more relevant for herds kept on grooved concrete and/or in permanent housing. In addition, sole thickness should be evaluated as a risk factor, as too-small thickness can be caused by excessive abrasion of the sole horn, overtrimming, and long walking distances from the pasture to the milking parlour. This is often observed in large herds with over 1000 cows, where long walking distances are common [83].

For welfare and recovery reasons, it is advisable to trim the lesion promptly and correctly and to apply an orthopaedic foot block under the healthy claw of the same leg. In many areas free from DD and *Treponema* spp. infections (e.g., in central and northern Europe), WLD and TTN lesions are frequently secondarily infected with pyrogenic bacteria, such as *Arcanobaterium pyogenes*. In areas with endemic DD-infected herds, investigations by both the Nottingham Lameness Expert Group and the University of Malaysia have shown that the presence of WLD in zone 3 is associated with a reduced likelihood of recovery, with cows that have been severely lame for a long time [96]. Regarding therapy, the current opinion is that an additional parenteral injection with an NSAID (e.g., an injection of Ketofen 10% solution at 3 mg/kg IM) results in even better recovery [86]. Herd veterinarians may choose another registered NSAID with a longer duration of action and possibly a more positive result, but this often also has consequences for the meat withdrawal period [97]. However, with the current legislation and considerations of animal welfare, farmers cannot cull such animals [98]. The positive effect of the combination of a block and an NSAID on lameness was also found in a recent review of the literature from Asia [66].

This is the best approach for the majority of CHDLs; however, it is not sufficient for the treatment of “non-healing” WLD in zone 3 and TTN, especially in areas endemic for infections with *Treponema* causing DD. TTN is almost always related to a secondary disorder and infection of the tip of the pedal bone [28]. Microbial investigations conducted at Liverpool University using PCR showed that, although the clinical and pathological presentations differ [78], *Treponema* bacteria are involved in both TTN and “non-healing” WLD [28]. TTN is, in our opinion, an osteitis of the pedal bone (P3), while “non-healing” WLD in zone 3 is a consequence of an infection of the pododerma of the hoof wall [78]. In both cases (non-healing white line lesions and TTN), the involvement of *Arcanobaterium pyogenes* and/or *Treponema* spp. in DD-endemic areas was proven [28], and tilmycosin treatment was found to have a positive effect. This research showed that a single treatment was always combined with surgical debridement of the infected area or resection of the infected tip of the pedal bone, of course, both under anaesthesia, and with an NSAID and a block under the contralateral claw. In cases of such complicated hoof lesions, this approach showed a very positive long-lasting effect. This treatment was also remarkable in cows affected by TTN or “non-healing” WLD in zone 3 for more than a year, who showed complete recovery. For farmers, the most important points were that the cows did not need attention after the removal of the foot block, and milk production was almost completely restored [67]. Tilmycosin, an antibiotic with a small molecular size, which is necessary for penetration into bone tissue, has been officially registered for use in cattle and claw disorders. Its therapeutic effect may be the consequence of a combination of antibacterial activity and an exaggerated immune response, such as that implicated in severe inflammatory reactions [99], which is responsible for the positive effect [67]. From a welfare perspective, it is irresponsible to send such lame cows to slaughter. The relative disadvantage of this approach is the additional use of antibiotics and the long withdrawal time for milk (35 days); therefore, it is advisable to use it restrictively. In the case of serious infection of the tip of the pedal bone, alternative treatments for TTN are, e.g., the removal of necrotic tissue with a grinder and partial amputation of the claw, both of which are performed under local anaesthesia with good pain management and with limited withdrawal times for milk and slaughter [88,89]. It is the responsibility of herd veterinarians to provide dairy farmers with the right information so that they can make a good decision together.

This study can be summarised as follows:

Prevention: The prevention of CHDLs starts with regular locomotion scoring at two-week intervals to detect WLD and TTN as early as possible [100,101], and, specifically, to detect WLD in zone 3 and TTN in cattle. Attention must be paid to proper preventive measures, including claw trimming; nutrition, for example, by employing the body condition score and achieving good ruminal function and a good mineral supply; and environmental housing management.

Claw Trimming: Research from Sweden indicates that, if CHDLs are the primary cause of lameness, strategic claw trimming—targeting lame cows, cows at the start of the dry standing period, and cows around three months in milk—is more effective than trimming all cows twice a year [91]. This strategy aligns with the greatest risk period for these disorders, which is between one and three months postpartum [92].

Nutrition: Good nutrition is crucial for preventing CHDLs, primarily by maintaining optimal rumen function. This promotes maximal endogenous biotin production; ensures an adequate supply of essential minerals, such as zinc, copper, manganese, and selenium [68,71,72]; and reduces the risk of sinking of the pedal bone within the hoof capsule, which subsequently leads to the development of CHDLs [18].

Environmental Management: Effective housing management to prevent CHDLs involves ensuring good cow comfort. This includes preventing overcrowding, properly adjusting cubicles, using rubber-coated floors when possible [73], and maintaining a clean and dry environment to prevent further contamination and promote healing. Cow comfort can be monitored using the cow standing index, as proposed by Cook et al. [93].

Veterinary Consultation: It is important to consult a bovine veterinarian for the appropriate treatment of complicated CHDLs. Veterinarians can provide guidance, perform underfoot anaesthesia, and intervene in cases of severe conditions or complications such as deep infections. This approach aims to prevent long-lasting lameness, improve cattle welfare and longevity, and enhance job satisfaction for dairy farmers and bovine practitioners.

## Data Availability

No new data were created or analyzed in this study. Data sharing is not applicable.

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
