# Peer review of "Review of White Line Disorders in Zone 3 and Toe Tip Necrosis in Dairy Cows and Recent Insights into Aetiopathogenesis and Treatments"

_microorganisms, 2025, doi:10.3390/microorganisms13092159_

Round 1

Reviewer 1 Report

Comments and Suggestions for Authors

The manuscript titled "White Line Disorders and Toe-Tip Necrosis in Dairy Cows, Re- 2
cent Insights into Aetiopathogenesis and Treatments" addresses an important topic in bovine medicine. However, some comments are to be addressed

1- The item "systematic review" should be added to the title

2-  The most important data  regarding pathogenesis and treatment should be added in the abstract

3- The limitations of your study should be added to the discussion section

4- The inclusion and exclusion criteria should  be added 

5- The limitation of using limited databases should be explained

6- Line 91: The expression "potentially interesting papers" what is the criteria of potentially interesting paper? is the journal impact factor? is the topic? Please clarify

Line 102: authors mentioned that Two review articles were included. To have a reliable conclusion, reviews should be deleted from the list.

Author Response

The manuscript titled "White Line Disorders and Toe-Tip Necrosis in Dairy Cows, Recent Insights into Aetiopathogenesis and Treatments" addresses an important topic in bovine medicine.

Dear Reviewers: Thank you very much for your time and all you critical comments and recommendations. As you may expect, all are evaluated carefully and to be honest it has resulted into a better manuscript.

Thank you,

Menno Holzhauer, 1st author

Royal GD Animal Health

1- The item "systematic review" should be added to the title

Au.: I agreed with you and followed your advice, but this was not according the Editorial guidelines, according these guidelines it must be Review.

2-  The most important data regarding pathogenesis and treatment should be added in the abstract

Au: we have adapted according your suggestion.

3- The limitations of your study should be added to the discussion section

Au.: this was added at the start of the discussion

4- The inclusion and exclusion criteria should be added 

Au: the inclusion and the exclusion criteria were mentioned in the objectives of this review

5- The limitation of using limited databases should be explained

Au: this review was limited to research papers published after the period of new insights into the aetio-pathogenesis of this type of disorders (around the year 2000). Research from before the turn of the century was focussed too much on the consequences of rumen acidosis. A statement was included in the paper.

6- Line 91: The expression "potentially interesting papers" what is the criteria of potentially interesting paper? is the journal impact factor? is the topic? Please clarify

Au: meant is here that are coming up after the database with the topic claw-horn lesions, white line and toe necrosis or toe tip necrosis

Line 102: authors mentioned that Two review articles were included. To have a reliable conclusion, reviews should be deleted from the list.

Au: yes we agree and have diluted the review articles from the table

Reviewer 2 Report

Comments and Suggestions for Authors

microorganisms-3847625__REFEREEs REPORT Aug 22-2025 

White Line Disorders and Toe-Tip Necrosis in Dairy Cows, Recent Insights into Aetiopathogenesis and Treatments.

 REMARKS AND RECOMMENDATIONS

The present study aims to clarify the pathogenesis of white line disease (WLD) and its associated claw lesions such as toe-tip necrosis (TTN), and to discuss practical treatment applications.

I have included several comments and questions in the report below that should be used to enhance the review article. These comments will help to expand on the content of this review paper, eliminate various ambiguities that run throughout the text, and clarify that in many countries not only Treponema spp. are involved in WLD and TTN infections.

TITLE: When reading the manuscript, it is not entirely clear whether the authors are referring solely to WLD at the toe and TTN, or also to WLD at zone 3 (which is the abaxial rear part of the white line area) of the 10 claw zones, where WLD most frequently occurs. Please ensure clarity in the title.

ABSTRACT

Line 12-13: “ …white line disease and toe-tip necrosis—often initiated as axial white line lesions— …”; When reading the manuscript, it is not entirely clear whether the authors are referring solely to WLD at the toe and TTN, or also to WLD at zone 3 (which is the abaxial rear part of the white line area) of the 10 claw zones, where WLD most frequently occurs. Please ensure clarity in the abstract, and throughout the entire manuscript.

Line 13: “…standard treatments ...”; You should explain what "Standard" treatments are. The challenge in choosing the correct treatment lies in making a comprehensive and definitive diagnosis, which is often overlooked when hoof trimmers or farmers treat lame cows, as is the case in many countries. We as veterinarians, we have learned to provide a definitive diagnosis and apply evidence-based treatments using "Standard operating procedures” for the proper treatment of claw disorders in cattle, as described recently. These aspects of making a correct diagnosis and selecting the appropriate treatment based on that diagnosis should be included in this review paper.

INTRODUCTION

Line 44: “…"Treponema spp."; "Treponema spp." should be italicized.

Line 55-57: “Other frequently mentioned risk factors for claw-horn disruption lesions (CHDL) include diseases related to fatty liver, such as mastitis and endometritis, nutrition (e.g., barley grain, protein, and fibre), housing and feeding management,…”; You should emphasize more strongly here that CHDL are closely related to laminitis, as numerous studies have reported (Machado et al. 2020; Foditsch et al. 2016; Griffiths et al. 2020 …). Additionally, you should include a reference regarding severe acute mastitis and endometritis as causes of CHDL in dairy cows. The two references Vermunt & Greenough (1995) and Bergsten (2003) do not encompass all of these mentioned relationships.

Line 63-65: “About fifteen years ago, a study conducted in Liverpool identified an association between Treponema bacteria involved in digital dermatitis (DD) and three 'non-healing' claw horn lesions: toe tip necrosis (TTN), 'non-healing' (NH-WLD), and non-healing sole ulcers.”; I am sorry, but Treponema spp. are not the only cause of infection at the white line and toe tip dermis. This depends on the digital dermatitis status of the herd (endemic infection or DD-free); Pyogenic bacteria also play also a major role, especially in DD-free herds, which can still be found in many Central European countries and some parts of Scandinavian countries. Please include this important information. Not all countries in the world have DD herd prevalences of over 90%, and therefore still have a significant proportion of DD-free herds.

Line 72: “…for various presentations of WLD and TTN in dairy cattle.”; This manuscript does not sufficiently define and describe the "various presentations of WLD and TTN" are not sufficiently defined and described in the manuscript. However, this should be addressed to provide clarity for readers.

Table 1: The authors mistakenly classified some studies in Table 1 as "experimental," which is inaccurate. For instance, Van Amstel et al.'s (2004) study was an observational cohort study, Newsome et al. 's (2016) study was a retrospective cohort study, Kofler's (2017) study was a review paper, and Somers et al. 's (2019) study is a prospective observational study, etc.. Please review all classifications in table 1 for accuracy!

PATHOGENESIS

Line 111: This chapter appears to be rather short to me, especially considering the authors' title mentions "Aetiopathogenesis of WLD and TTN." It should provide more detailed information on the possible causes. Currently, the information is somewhat superficial, which is not suitable for a review paper. Additionally, the significant impact of bovine laminitis on the development of WLD and other laminitis-related claw horn disruption lesions (such as double soles, sole ulcers, etc.) is missing in this chapter on aetiopathogenesis. References by Greenough (2007), Offer et al. (2003), Offer et al. (2004), Shearer & Van Amstel (2017), Tarlton et al. (2002), Machado et al. (2010), Foditsch et al. (2016), Griffiths et al. (2020) … should be consulted for further information. The focus in the aetiopathogenesis chapter seems to be directed towards the role of treponemes. However, infection by treponemes is always a secondary issue once CHDL has developed. This is either laminitis-related or traumatic due to factors related to the abrasiveness of the floor, extended standing times (overcrowding, capacity of the milking system, quality and hardness of the cubicles, heat stress …), long walking distances, excessive hoof trimming, etc.

Line 115-116: “… with white line separation [16]. probably the most….; The period after the reference appears to be incorrect.

Line 119: “These infections ….”; Please, some information here to confirm if all feedlots have an endemic DD infection. Without an endemic DD- infection, we would mainly expect infections with pyogenic bacteria.

Line 122: “In Western Europe we do not have feedlot cattle, but we see the same problems with proven Treponema infections in dairy herds ….”; I am sorry, but infections caused by Treponema spp. are not the only factor to consider. The DD status of the herd plays a significant role, whether they are endemically infected or DD-free. It is important to note that not all dairy herds in Europe have endemic DD infections; in many cases, pyogenic bacteria play also a significant role. Could you please provide more specific details about this situation and rephrase these sentences.

Line 126-127: ”In line with a Canadian study, our experience is that TTN starts with a disorder of/separation of the axial white line, …”; However, the authors do not specify in this chapter where exactly on the claw WLD typically occurs? Is it abaxial in the posterior region of the white line (zone 3)? This information should also be clearly stated here.

Line 129: “…housing factors like grooved slats, sharp turns and high stocking density”; I highly suggest adding other important risk factors here, such as overtrimming by inexperienced hoof trimmers (refer to references on overtrimming or excessive trimming).

Line 130: “Research showed less CHD in free stalls [37] and on rubber-covered slatted floors [38-40]; The comparison group is missing in this sentence. Are you trying to compare it to tie-stalls? According to my literature research, this statement is not accurate. In loose housing, we have significantly more WLD than in tied housing, but the prevalence may be lower in loose housing with rubber mats than in loose housing with concrete floors (refer to Jewell et al. (2019): Prevalence of lameness and associated risk factors on dairy farms in the Maritime Provinces of Canada and numerous other articles on lameness and risk factors of claw lesions in tie stalls). Please revise for clarity and rephrase.

TREATMENT

Line 136-137: “After these steps one can make their conclusion that a white-line disorder is the most probable cause of lameness...”; Please note that the operator must use the hoof tester to assess if the visible WLD (white line separation ...) is associated with any pain. Without evaluation for a painful reaction, other causes of lameness, including those from proximal limb regions, cannot be ruled out (please reference this sentence).

Line 137-138: “…Many people and the Nottingham Research Group have found a positive effect with the use …”; Here you should add at least a second reference (for other people). I agree with this statement, but only in cases where the infection is limited to the dermis, but not for cases with deep digital sepsis. Please, specify this distinction.

Line 142, Figure 1: Can you please clarify what you mean by WLD grade 1? Are you referring to white line separation without dermis infection, or a white line infection (white line abscess) affecting a small area of the dermis, approximately 1 cm2 in size?

Line 162: “If the local infection is more extensive …”; Please clarify what you mean by "more extensive"? Are you referring to infections of the dermis at the white line (= white line abscess)? and non-healing white-line abscesses, or cases involving infection of the adjoining bone as well?

Line 163: “…and it often becomes complicated, …”; Please clarify what you mean by "complicated"? Are you referring to extensive secondary DD-infection of the dermis or a bone infection? Please specify.

Line 171, Figure 2: In reference to the presented figure 2 C, I recommend adding to the legend that “surgical debridement” (as shown in this figure), should always be performed using local anesthesia.

Line 186-187: “Infections may penetrate the corium and progress to P3 osteitis, P2 osteomyelitis, tendonitis, tenosynovitis, cellulitis, …”: Please, add here information about regarding the most common bacteria found in P3 osteitis.

Line 197: “Removal of the necrotic tissue by surgical intervention or amputation of the tip of the claw [47]”; Please, add “using local anesthesia” here.

PREVENTION

One important preventive measure is missing here: regular locomotion scoring of the herd at two-weeks intervals to be able to detect lameness and WLD lesions early, perhaps before the secondary infection of the corium by Treponema spp.. Please, add appropriate references for this important preventive measure.

Line 205-206: “…it is necessary to pay attention to proper preventive claw trimming, nutrition and the environment..”: Please specify what one should expect with "Proper preventive trimming": correct dorsal wall length, sole horn thickness of at least 5-7 mm, correct heel height especially of the medial hind claw, and proper removal of load from a claw affected by WLD separation or other CHDL (step 4 of functional hoof trimming).

DISCUSSION

Line 224: Please clarify in the discussion that not all WLD and TTN lesions are secondarily infected with Treponema spp. This is particularly relevant in herds with endemic digital dermatitis (DD)-infection, which is present in more than 90% of herds in many countries. However, there are also countries in Central and Northern Europe where "only" approximately 50% of dairy herds are currently endemic with DD, and the remaining herds are DD-free. In DD-free herds, Treponema spp. plays no role, but mainly pyogenic bacteria, which unfortunately often lead to infection of the pedal bone and later the distal interphalangeal joint. Please discuss this aspect in your study, as it is a review study intended to encompass situations beyond those in your country and other countries with more than 90% DD herd prevalence.

Line 226: “White line disorders are one of the most prevalent CHD lesions in dairy cattle in Western Europe and the USA [13,52].”; Reference 52 is not an original literature; it is a review paper. Please cite a recent and original article reporting a high prevalence/incidence of WLD/WLA in cows in loose housing systems.

Line 231: “…are a consequence of a combination of poorer horn quality …”; Please also include THIN SOLES as a risk factor. Thin soles can be caused by excessive abrasion of the sole horn, overtrimming, and long walking distances from pasture to the milking parlor. This is often observed in large herds with over 1000 cows, where such long walking distances are common.

Line 246: “This approach will the best the best advice for the majority of the CHDL, but this is not sufficient for the treatment of ‘non-healing’ WLD and TTN.”; Yes, I agree with your differentiation. However, I suggest adding some explanatory information. In Central Europe, we still have many herds (approx. 50%) that are free from DD-infection.

Line 251: “TTN is in our opinion an osteitis of the pedal bone (P3) …”; I am not aware of any publication that describes cases of infection of the tip of the pedal bone with Treponema spp. as the causative bacteria. According to the literature, infections of the pedal bone are primarily caused by pyogenic bacteria. Please discuss this aspect further here.

Line 254: “In both cases (non-heling white line lesions and TTN)…”; … non-healing …

Line 255: “…a positive effect was found with treatment with tilmycosin”; Please differentiate between the treatment of an infected dermis at the toe and an infected bone. In the latter case, veterinarians should always perform a surgical debridement or resection of the infected tip of the pedal bone, rather than relying solely on systemic antibiotics. Please provide clarification on this point.

Line 267-268: “An alternative for TTN is e.g., removal of the necrotic tissue with a grinder or partial amputation of the claw…”; Based on the relevant literature, surgery is not an alternative but rather the only effective treatment method when dealing with a bone infection. Once again, we encounter the issue of a definitive diagnosis: does TTN always indicate an infection of the tip of the pedal bone, or can it also refer to an infection solely in the dermis at the tip of the pedal bone? Please provide clarification and rephrase this sentence.

Line 272: Prevention of claw health disorders (CHD) …; Please add regular locomotion scoring at two-week intervals to detect WLD and TTN as early as possible. Additionally, please include appropriate references such as ICAR 2022: ICAR Guidelines: Section 7 - Functional traits in dairy cattle - https://www.icar.org/Guidelines/07-Bovine-Functional-Traits.pdf; and/or Eriksson HK, Daros RR, Von Keyserlingk MAG, Weary DM (2020): Effects of case definition and assessment frequency on lameness incidence estimates. J Dairy Sci 103:638–648).

Line 281: “Nutrition: Good nutrition is crucial for preventing CHD lesions, primarily through maintaining optimal rumen function. This promotes maximal endogenous biotin production and ensures an adequate supply of essential minerals such as zinc, copper, manganese, and selenium [35,36,45].” Here you should also add, ...."and reduces the risk of sinking of the pedal bone within the hoof capsule, which subsequently leads to the development of CHDL. Please include relevant literature references.

Thank you.

Comments on the Quality of English Language

I do not have the necessary expertise to judge this.

Author Response

 REMARKS AND RECOMMENDATIONS REVIEWER

The present study aims to clarify the pathogenesis of white line disease (WLD) and its associated claw lesions such as toe-tip necrosis (TTN), and to discuss practical treatment applications.

I have included several comments and questions in the report below that should be used to enhance the review article. These comments will help to expand on the content of this review paper, eliminate various ambiguities that run throughout the text, and clarify that in many countries not only Treponema spp. are involved in WLD and TTN infections.

Dear Reviewer: Thank you very much for your time and all you recommendations. As you may expect, all are evaluated carefully and to be honest it has resulted into a better manuscript.

TITLE: When reading the manuscript, it is not entirely clear whether the authors are referring solely to WLD at the toe and TTN, or also to WLD at zone 3 (which is the abaxial rear part of the white line area) of the 10 claw zones, where WLD most frequently occurs. Please ensure clarity in the title.

Au.: we have adapted the title according your suggestions. In literature different classifications for the zones are used and we now have used the classification as proposed in the paper of van der Tol et al., 2002.

ABSTRACT

Line 12-13: “ …white line disease and toe-tip necrosis—often initiated as axial white line lesions— …”; When reading the manuscript, it is not entirely clear whether the authors are referring solely to WLD at the toe and TTN, or also to WLD at zone 3 (which is the abaxial rear part of the white line area) of the 10 claw zones, where WLD most frequently occurs. Please ensure clarity in the abstract, and throughout the entire manuscript.

Au.: adapted according your suggestion, we followed here the classification as presented by van der Tol et al., 2002. This classification of the different zones is presented differently by different authors. 

Line 13: “…standard treatments ...”; You should explain what "Standard" treatments are. The challenge in choosing the correct treatment lies in making a comprehensive and definitive diagnosis, which is often overlooked when hoof trimmers or farmers treat lame cows, as is the case in many countries. We as veterinarians, we have learned to provide a definitive diagnosis and apply evidence-based treatments using "Standard operating procedures” for the proper treatment of claw disorders in cattle, as described recently. These aspects of making a correct diagnosis and selecting the appropriate treatment based on that diagnosis should be included in this review paper.

Au.: adapted here and in line 127 according your suggestion.

 INTRODUCTION

Line 44: “…"Treponema spp."; "Treponema spp." should be italicized.

Au.: done

Line 55-57: “Other frequently mentioned risk factors for claw-horn disruption lesions (CHDL) include diseases related to fatty liver, such as mastitis and endometritis, nutrition (e.g., barley grain, protein, and fibre), housing and feeding management,…”; You should emphasize more strongly here that CHDL are closely related to laminitis, as numerous studies have reported (Machado et al. 2020; Foditsch et al. 2016; Griffiths et al. 2020 …). Additionally, you should include a reference regarding severe acute mastitis and endometritis as causes of CHDL in dairy cows. The two references Vermunt & Greenough (1995) and Bergsten (2003) do not encompass all of these mentioned relationships.

Au: we have adapted to your suggestion and agreed that both publications we mentioned not completely cover the problems and have added Ingvartsen, 2004 who described the consequences of fatty liver problems

Line 63-65: “About fifteen years ago, a study conducted in Liverpool identified an association between Treponema bacteria involved in digital dermatitis (DD) and three 'non-healing' claw horn lesions: toe tip necrosis (TTN), 'non-healing' (NH-WLD), and non-healing sole ulcers.”; I am sorry, but Treponema spp. are not the only cause of infection at the white line and toe tip dermis. This depends on the digital dermatitis status of the herd (endemic infection or DD-free); Pyogenic bacteria also play also a major role, especially in DD-free herds, which can still be found in many Central European countries and some parts of Scandinavian countries. Please include this important information. Not all countries in the world have DD herd prevalences of over 90%, and therefore still have a significant proportion of DD-free herds.

Au: again we agree that not all these problems are a consequence of Treponema spp., but several investigations have shown that in case of ‘non-healing’ disorders Treponema spp. were found consistently as the most probably causing organism. Our experience was that these herds did not always have more problems with DD but indeed in areas like Sweden where a lot of herds are (almost) free of DD and they have not these claw-horn lesions. So adapted.

Line 72: “…for various presentations of WLD and TTN in dairy cattle.”; This manuscript does not sufficiently define and describe the "various presentations of WLD and TTN" are not sufficiently defined and described in the manuscript. However, this should be addressed to provide clarity for readers.

Au: adapted according your suggestion

Table 1: The authors mistakenly classified some studies in Table 1 as "experimental," which is inaccurate. For instance, Van Amstel et al.'s (2004) study was an observational cohort study, Newsome et al. 's (2016) study was a retrospective cohort study, Kofler's (2017) study was a review paper, and Somers et al. 's (2019) study is a prospective observational study, etc.. Please review all classifications in table 1 for accuracy!

Au: we serious evaluated Table 1 and discussed within the epidemiology group, but this is not always simple, e.g. in heritability studies you score at the moment of claw trimming (at random) and afterwards you collect data about the sires at the breeding organisations. According the epidemiologist is this case-control. For example you have cases (e.g. sole ulcer present) and others have not (control)> that is how we now have evaluated this table.   

PATHOGENESIS

Line 111: This chapter appears to be rather short to me, especially considering the authors' title mentions "Aetiopathogenesis of WLD and TTN." It should provide more detailed information on the possible causes. Currently, the information is somewhat superficial, which is not suitable for a review paper. Additionally, the significant impact of bovine laminitis on the development of WLD and other laminitis-related claw horn disruption lesions (such as double soles, sole ulcers, etc.) is missing in this chapter on aetiopathogenesis. References by Greenough (2007), Offer et al. (2003), Offer et al. (2004), Shearer & Van Amstel (2017), Tarlton et al. (2002), Machado et al. (2010), Foditsch et al. (2016), Griffiths et al. (2020) … should be consulted for further information. The focus in the aetiopathogenesis chapter seems to be directed towards the role of treponemes. However, infection by treponemes is always a secondary issue once CHDL has developed. This is either laminitis-related or traumatic due to factors related to the abrasiveness of the floor, extended standing times (overcrowding, capacity of the milking system, quality and hardness of the cubicles, heat stress …), long walking distances, excessive hoof trimming, etc.

Au: this was serious adapted according your suggestion

Line 115-116: “… with white line separation [16]. probably the most….; The period after the reference appears to be incorrect.

Au: adapted, period removed

Line 119: “These infections ….”; Please, some information here to confirm if all feedlots have an endemic DD infection. Without an endemic DD- infection, we would mainly expect infections with pyogenic bacteria.

Au: adapted according your suggestion

Line 122: “In Western Europe we do not have feedlot cattle, but we see the same problems with proven Treponema infections in dairy herds ….”; I am sorry, but infections caused by Treponema spp. are not the only factor to consider. The DD status of the herd plays a significant role, whether they are endemically infected or DD-free. It is important to note that not all dairy herds in Europe have endemic DD infections; in many cases, pyogenic bacteria play also a significant role. Could you please provide more specific details about this situation and rephrase these sentences.

Au: sentences are rephrased according your suggestion.

Line 126-127: ”In line with a Canadian study, our experience is that TTN starts with a disorder of/separation of the axial white line, …”; However, the authors do not specify in this chapter where exactly on the claw WLD typically occurs? Is it abaxial in the posterior region of the white line (zone 3)? This information should also be clearly stated here.

Au: extra information added

Line 129: “…housing factors like grooved slats, sharp turns and high stocking density”; I highly suggest adding other important risk factors here, such as overtrimming by inexperienced hoof trimmers (refer to references on overtrimming or excessive trimming).

Au: adapted according your suggestion

Line 130: “Research showed less CHD in free stalls [37] and on rubber-covered slatted floors [38-40]; The comparison group is missing in this sentence. Are you trying to compare it to tie-stalls? According to my literature research, this statement is not accurate. In loose housing, we have significantly more WLD than in tied housing, but the prevalence may be lower in loose housing with rubber mats than in loose housing with concrete floors (refer to Jewell et al. (2019): Prevalence of lameness and associated risk factors on dairy farms in the Maritime Provinces of Canada and numerous other articles on lameness and risk factors of claw lesions in tie stalls). Please revise for clarity and rephrase.

Au: Clarified and revised

TREATMENT

Line 136-137: “After these steps one can make their conclusion that a white-line disorder is the most probable cause of lameness...”; Please note that the operator must use the hoof tester to assess if the visible WLD (white line separation ...) is associated with any pain. Without evaluation for a painful reaction, other causes of lameness, including those from proximal limb regions, cannot be ruled out (please reference this sentence).

Au: adapted according your suggestion

Line 137-138: “…Many people and the Nottingham Research Group have found a positive effect with the use …”; Here you should add at least a second reference (for other people). I agree with this statement, but only in cases where the infection is limited to the dermis, but not for cases with deep digital sepsis. Please, specify this distinction.

Au.: adapted according your suggestion

Line 142, Figure 1: Can you please clarify what you mean by WLD grade 1? Are you referring to white line separation without dermis infection, or a white line infection (white line abscess) affecting a small area of the dermis, approximately 1 cm2 in size?

Au: we followed here the graduation as described in the ICAR health atlas (added in the paper). So here we have indeed a white line infection affecting a small area of the dermis. 

Line 162: “If the local infection is more extensive …”; Please clarify what you mean by "more extensive"? Are you referring to infections of the dermis at the white line (= white line abscess)? and non-healing white-line abscesses, or cases involving infection of the adjoining bone as well?

Au.: clarified in the text

Line 163: “…and it often becomes complicated, …”; Please clarify what you mean by "complicated"? Are you referring to extensive secondary DD-infection of the dermis or a bone infection? Please specify.

Au.: I hope we have clarified this according your request. Normally open the lesion, removing the affected horn and a block and NSAID is enough to solve the problem  

Line 171, Figure 2: In reference to the presented figure 2 C, I recommend adding to the legend that “surgical debridement” (as shown in this figure), should always be performed using local anesthesia.

Au.: we agree and have added that in the paper

Line 186-187: “Infections may penetrate the corium and progress to P3 osteitis, P2 osteomyelitis, tendonitis, tenosynovitis, cellulitis, …”: Please, add here information about regarding the most common bacteria found in P3 osteitis.

Au: the most frequently found pathogen are E. coli and Trueperella pyogenes, added in the paper

Line 197: “Removal of the necrotic tissue by surgical intervention or amputation of the tip of the claw [47]”; Please, add “using local anesthesia” here.

Au.: added in the paper

PREVENTION

One important preventive measure is missing here: regular locomotion scoring of the herd at two-weeks intervals to be able to detect lameness and WLD lesions early, perhaps before the secondary infection of the corium by Treponema spp.. Please, add appropriate references for this important preventive measure.

Au.: thank you again, this was added to the paper

Line 205-206: “…it is necessary to pay attention to proper preventive claw trimming, nutrition and the environment..”: Please specify what one should expect with "Proper preventive trimming": correct dorsal wall length, sole horn thickness of at least 5-7 mm, correct heel height especially of the medial hind claw, and proper removal of load from a claw affected by WLD separation or other CHDL (step 4 of functional hoof trimming).

Au: adapted according your suggestion

 DISCUSSION

Line 224: Please clarify in the discussion that not all WLD and TTN lesions are secondarily infected with Treponema spp. This is particularly relevant in herds with endemic digital dermatitis (DD)-infection, which is present in more than 90% of herds in many countries. However, there are also countries in Central and Northern Europe where "only" approximately 50% of dairy herds are currently endemic with DD, and the remaining herds are DD-free. In DD-free herds, Treponema spp. plays no role, but mainly pyogenic bacteria, which unfortunately often lead to infection of the pedal bone and later the distal interphalangeal joint. Please discuss this aspect in your study, as it is a review study intended to encompass situations beyond those in your country and other countries with more than 90% DD herd prevalence.

Au we agree and adapted that in the paper

Line 226: “White line disorders are one of the most prevalent CHD lesions in dairy cattle in Western Europe and the USA [13,52].”; Reference 52 is not an original literature; it is a review paper. Please cite a recent and original article reporting a high prevalence/incidence of WLD/WLA in cows in loose housing systems.

Au: we have adapted the reference into Jewell et al., 2018

Line 231: “…are a consequence of a combination of poorer horn quality …”; Please also include THIN SOLES as a risk factor. Thin soles can be caused by excessive abrasion of the sole horn, overtrimming, and long walking distances from pasture to the milking parlor. This is often observed in large herds with over 1000 cows, where such long walking distances are common.

Au: adapted in the paper

Line 246: “This approach will the best the best advice for the majority of the CHDL, but this is not sufficient for the treatment of ‘non-healing’ WLD and TTN.”; Yes, I agree with your differentiation. However, I suggest adding some explanatory information. In Central Europe, we still have many herds (approx. 50%) that are free from DD-infection.

Au: adapted in the paper

Line 251: “TTN is in our opinion an osteitis of the pedal bone (P3) …”; I am not aware of any publication that describes cases of infection of the tip of the pedal bone with Treponema spp. as the causative bacteria. According to the literature, infections of the pedal bone are primarily caused by pyogenic bacteria. Please discuss this aspect further here.

Au: in the paper of Evans et al., 2011 Treponema spp. was approved in different types of “non-healing” claw-horn lesions like TN etc. Adaption in the paper  

Line 254: “In both cases (non-heling white line lesions and TTN)…”; … non-healing …

Au: adapted

Line 255: “…a positive effect was found with treatment with tilmycosin”; Please differentiate between the treatment of an infected dermis at the toe and an infected bone. In the latter case, veterinarians should always perform a surgical debridement or resection of the infected tip of the pedal bone, rather than relying solely on systemic antibiotics. Please provide clarification on this point.

Au: we agree and have adapted in the paper

Line 267-268: “An alternative for TTN is e.g., removal of the necrotic tissue with a grinder or partial amputation of the claw…”; Based on the relevant literature, surgery is not an alternative but rather the only effective treatment method when dealing with a bone infection. Once again, we encounter the issue of a definitive diagnosis: does TTN always indicate an infection of the tip of the pedal bone, or can it also refer to an infection solely in the dermis at the tip of the pedal bone? Please provide clarification and rephrase this sentence.

Au: I (MH) have present personally different of such cases of TN to the pathologist and in 100% of those cases, he concluded that this was an infectious osteitis at the tip of the pedal bone. With another pathologists I had discussions about overtrimming but frequently heifers presented to the pathologist were never trimmed before the lesion was diagnosed (but that is not your point). In the study performed in 2024, debridement under anaesthesia was performed in combination with local and parenteral treatment. This was performed at patients that were already treated different times before locally + block + NSAID without satisfying result.  

Line 272: Prevention of claw health disorders (CHD) …; Please add regular locomotion scoring at two-week intervals to detect WLD and TTN as early as possible. Additionally, please include appropriate references such as ICAR 2022: ICAR Guidelines: Section 7 - Functional traits in dairy cattle - https://www.icar.org/Guidelines/07-Bovine-Functional-Traits.pdf; and/or Eriksson HK, Daros RR, Von Keyserlingk MAG, Weary DM (2020): Effects of case definition and assessment frequency on lameness incidence estimates. J Dairy Sci 103:638–648).

Au: adapted in the paper

Line 281: “Nutrition: Good nutrition is crucial for preventing CHD lesions, primarily through maintaining optimal rumen function. This promotes maximal endogenous biotin production and ensures an adequate supply of essential minerals such as zinc, copper, manganese, and selenium [35,36,45].” Here you should also add, ...."and reduces the risk of sinking of the pedal bone within the hoof capsule, which subsequently leads to the development of CHDL. Please include relevant literature references.

Au: adapted in the paper references included.

Round 2

Reviewer 2 Report

Comments and Suggestions for Authors

 REMARKS AND RECOMMENDATIONS

The authors have incorporated all questions and comments into the revised version of the manuscript. However, the revised version contains new minor errors that need to be corrected.

Line 77: pyogenic instead of “pyrogenc”.

Table 1: The font type and font size in table 1 is completely different from the text!?

Lines 167, 168, 175-178, 2023-204, 263: The font type and font size in these lines is completely different from the other text!?

Line 169: [Cramer and Solano Merck]. This citation is certainly not correct, please check.

Comments on the Quality of English Language

 REMARKS AND RECOMMENDATIONS

The authors have incorporated all questions and comments into the revised version of the manuscript. However, the revised version contains new minor errors that need to be corrected.

Line 77: pyogenic instead of “pyrogenc”.

Table 1: The font type and font size in table 1 is completely different from the text!?

Lines 167, 168, 175-178, 2023-204, 263: The font type and font size in these lines is completely different from the other text!?

Line 169: [Cramer and Solano Merck]. This citation is certainly not correct, please check.

Author Response

Dear Reviewers: Thank you very much for your time and all you critical comments and recommendations. Based on your advices the paper was also checked for the correct grammar by the internal professional editing Services of MDPI

As you may expect, all are evaluated carefully and to be honest it has resulted into a better manuscript. Do not hesitate if you have any questions.

Thank you,

Menno Holzhauer, 1st author

Royal GD Animal Health

Comments Reviewer 2

Line 77: pyogenic instead of “pyrogenc”.

Au: adapted

Table 1: The font type and font size in table 1 is completely different from the text!?

Au: you are right and we have corrected

Lines 167, 168, 175-178, 2023-204, 263: The font type and font size in these lines is completely different from the other text!?

Au: we have checked and adapted this

Line 169: [Cramer and Solano Merck]. This citation is certainly not correct, please check.

Au: reference is ok now